# DynamicKV: Task-Aware Adaptive KV Cache Compression for Long Context LLMs

## Abstract

Efficiently managing the KV cache in Large Language Models (LLMs) is a critical challenge for long-context processing tasks such as retrieval-augmented generation (RAG), long text summarization, and multi-document analysis. Extending the context length substantially increases the KV cache size, leading to excessive memory consumption. Existing KV cache compression methods enforce a fixed pattern, neglecting task-specific characteristics, which hampers the effective retention of essential information while discarding less important tokens. In this paper, we introduce a novel Task-Aware KV cache mechanism that dynamically adjusts the KV cache size across different layers based on the characteristics of the tasks. Our approach builds on the significant observation of distinct activation patterns across layers in various tasks, which highlights the need for adaptive strategies tailored to each task's unique demands. Based on this insight, we propose DynamicKV, a method that dynamically optimizes token retention by adjusting the number of tokens retained at each layer, adapting to the specific task. DynamicKV establishes global and per-layer maximum KV cache budgets, temporarily retaining the maximum budget for the current layer, and periodically updating the KV cache sizes of all preceding layers during inference. Our method demonstrates exceptional performance on the LongBench dataset, retaining only 1.7% of the KV cache while preserving 90%, 87%, 78%, and 83% of the original accuracy for LlaMA-3-8B-Instruct, Mistral-7B-Instruct-v0.2, Qwen2-7B-Instruct, and InternLM-2.5-7B-Chat-1M, respectively. When the retained KV cache size is increased to 6.9%, the performance becomes nearly indistinguishable from that without any KV cache compression. Notably, even under extreme compression (0.9%), DynamicKV surpasses state-of-the-art (SOTA) methods by 11% in the Needle-in-a-Haystack test using Mistral-7B-Instruct-v0.2. The code will be released to the public.

## 1 Introduction

Large Language Models (LLMs) (Achiam et al., 2023; Radford, 2018; Radford et al., 2019) are exerting a considerable influence in the field of natural language processing (NLP), driving advancements in document summarization, content creation, code generation, and dialogue systems (Chiang et al., 2023). Recent developments in LLMs (Liu et al., 2024b) have been scaled up to handle long contexts, with LlaMA3 (Dubey et al., 2024) processing up to 32K tokens and InternLM (Cai et al., 2024) handling 1M tokens. However, scaling LLMs to handle extended contexts inherently incurs a substantial delay due to the quadratic complexity of attention mechanisms with increasing context length. A widely adopted solution to alleviate these delays is caching the key and value (KV) states of previous tokens (Waddington et al., 2013). Despite this optimization, handling long sequences still demands substantial memory (*e.g.,* maintaining a KV cache for 100K tokens in LlaMA2-7B (Touvron et al., 2023) consumes over 50GB of memory).

To address this issue, recent studies have explored the optimization of KV caching, including KV cache quantization (Kang et al., 2024; Hooper et al., 2024), token dropping (Zhang et al., 2024b; Xiao et al., 2023), architectural improvements to Transformers (Sun et al., 2024), KV cache fusion (Nawrot et al., 2024), and hierarchical sharing and constraints(Liu et al., 2024a; Brandon et al., 2024). Existing KV cache compression methods enforce a fixed pattern (as shown in Figure 1), such as a hierarchical pyramid structure (Zhang et al., 2024a) or a structure similar to FastGen's fixed

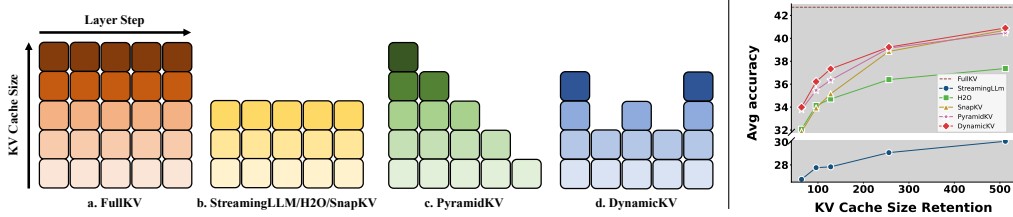

Figure 1: Comparison of DynamicKV with traditional methods in maintaining KV cache size across layers. Left: the structure difference: (a) Retain all KV cache. (b) Fixed KV cache for each layer (e.g., StreamingLLM, H2O, SnapKV). (c) Hierarchically decreasing pyramid KV cache retention. (d) Ours DynamicKV: layer-aware adaptive KV cache retention. Right: average accuracy on different KV cache rentention.

internal pattern (Ge et al., 2023), or they fix the length of the KV cache to selectively retain tokens across different layers (Zhang et al., 2024b; Li et al., 2024). However, LLMs require different numbers of layers when handling different types of tasks. For example, for knowledge-based question answering tasks, only the first few layers are needed to achieve high accuracy, while for complex reasoning tasks (*e.g.,* mathematics and code generation), more layers are often required to achieve higher accuracy (Elhoushi et al., 2024). Thus, we raise a question: ***Do different types of tasks all follow a fixed pattern?***

To examine this question, we aim to systematically investigate the design principles of the KV cache compression across different tasks. Inspired by Zhang et al. (2024a), we first investigate how information flow is aggregated through attention mechanisms across different layers in four types of tasks, including single- and multi-document QA, summarization, synthetic tasks and code completion. We find that the attention distribution varies for different types of tasks. For example, in summarization tasks, the upper layers require a small KV cache sizes, while code completion tasks need larger KV cache sizes in the upper layers. This implies that for code completion tasks, upper layers require maintaining a larger KV cache size, in contrast to PyramidKV (Zhang et al., 2024a), where the KV cache size decreases as the layer depth increases.

Building on this insight, we propose a task-aware adaptive KV cache compression method, named DynamicKV. Specifically, we first calculate an attention score for the most recent few tokens and all other tokens, which in RAG (Lewis et al., 2020) can be viewed as calculating the relevance of the most recent query to the retrieved text. Then, we preset a temporary storage to hold the temporary KV cache states, and gradually calculate the size of the final retained temporary storage at each k layer by calculating the size of the correlation mean. It should be noted that at each update, the value is gradually normalized, and the retained temporary storage at each layer is always smaller than the previous one. This temporary storage is determined by the number of tokens that need to be retained, and its size is much smaller than the original cache, thus imposing minimal memory overhead.

We validate our DynamicKV on 16 datasets from LongBench (Bai et al., 2023), demonstrating robust performance across multiple models, including LlaMA-3-8B-instruct (Dubey et al., 2024), qwen-2-7B-instruct (Yang et al., 2024), mistral-7b-chat-v0.2 (Jiang et al., 2023), internlm-2.5-7b-chat-1M (Cai et al., 2024). Our DynamicKV exhibit superior overall effectiveness compared to conventional fixed-pattern methods (Zhang et al., 2024b; Li et al., 2024; Nawrot et al., 2024). Notably, DynamicKV is able to retain full performance while utilizing only 6.9% of the tokens, and in extreme scenarios, it preserve 90% of the performance with just 1.7% of the tokens. Furthermore, experiments on the Needle in a Haystack benchmark revealed that DynamicKV significantly outperforms state-of-the-art (SOTA) methods.

## 2 RELATED WORK

**Potential patterns of attention.** The Transformer architecture (Vaswani, 2017) becomes a cornerstone in NLP by stacking multiple layers to progressively refine input data. BERT (Devlin, 2018), a model based on this architecture, (Jawahar et al., 2019)demonstrates that intermediate layers encode

a rich hierarchy of linguistic information: from surface-level features at the bottom, through syntactic features in the middle, to semantic features at the top. This indicates that models are capable not only of understanding lexical information but also of grasping more complex linguistic structures. For decoder-only LLMs, (Fan et al., 2024) observes that not all layers are necessary for simple tasks, as intermediate layers can often achieve comparable performance to the final layer. Techniques like (Elhoushi et al., 2024), which involve increasing dropout in lower layers during training, allow the model to exit computation early, reducing resource consumption. To optimize model inference efficiency, especially in terms of KV cache compression, (Brandon et al., 2024) proposes cross layer attention(CLA), which can reduce the KV cache size by at least half by sharing cross-layer attention, significantly lowering memory usage. Ada-KV (Feng et al., 2024b) visualizes attention distributions across all layers have also shown that attention patterns dynamically evolve as the layers progress. Inspired by these findings, we aims to dynamically select and adjust the number of tokens to retain per layer, combining inter-layer redundancy identification with efficient KV cache management. This approach aims to maintain high-quality output while improving inference efficiency.

**Token drop.** Token drop is a strategy designed to reduce memory usage by selectively retaining the most influential tokens in the KV cache during the inference phase of LLMs. Due to its plug-and-play nature, the token drop method can often be applied to different models without incurring any additional costs. FastGen (Ge et al., 2023) evicts unnecessary contexts and discards non-special tokens based on the recognized structure of attention modules by effectively analyzing the token information within attention patterns. Scissorhands (Liu et al., 2024c) exploits the hypothesis of the persistence of importance, suggesting that tokens with significant influence at one point will continue to impact future generations. By using attention scores as a metric and applying a Least Recently Used (LRU) cache eviction strategy, it discards non-critical tokens to optimize memory usage. StreamingLLM (Xiao et al., 2023) leverages the characteristics of attention sinks in LLMs to focus on streaming processing with dynamic adjustment of the KV cache. H2O (Zhang et al., 2024b) proposes a scoring function based on accumulated attention scores for greedily evicting KV pairs during generation. SnapKV (Li et al., 2024) primarily achieves compression by selectively targeting key positions for each attention head. PyramidKV (Zhang et al., 2024a) identified the phenomenon of massive activation and adopted a hierarchical structure to optimize the number of KV cache entries retained at each layer. Although the PyramidKV approach considers the varying information density across different layers, its pyramidal pattern does not generalize across multiple models or tasks. LazyLLM (Fu et al., 2024) utilizes dynamic token pruning and an Aux Cache mechanism, allowing the model to select different subsets of tokens from the context at various generation steps, even reviving tokens pruned in previous steps. Ada-KV (Feng et al., 2024a) breaks from the conventional approach of uniform budget allocation across attention heads within layers, optimizes the eviction loss upper bound, leading to improved performance under various memory constraints when integrated with SnapKV and PyramidKV.

## 3 OBSERVATION

To systematically investigate the attention mechanism across layers in LLMs for long-context inputs, we conduct a fine-grained analysis of four tasks: single- and multi-document question answering (QA), summarization, synthetic tasks, and code completion. The main target is to investigate the distribution of attention in these various tasks, thereby enhancing our understanding of how the model aggregates dispread information within long-context inputs to generate accurate responses.

In particular, we focus our analysis on LlaMA (Dubey et al., 2024), visualizing the distribution and behavior of attention across layers to gain deeper insights into its internal mechanisms. Inspired by Zhang et al. (2024a), we calculate the average attention scores between the most recent tokens and all other tokens. Based on these scores, we then identify the top-k (128 multiplied by the number of layers) tokens with the highest attention across all layers, resulting in a layer distribution map denoted as Figure 2.

We observe a significant drop in the KV cache size requirement at the lower layers across the four tasks, indicating that only a small KV cache is needed in these layers. In contrast, the upper layers show a clear upward trend, suggesting that larger KV cache sizes are necessary, particularly in the code completion task, where complex reasoning is required. This phenomenon underscores that tasks involving complex reasoning demand larger KV cache sizes in the upper layers.

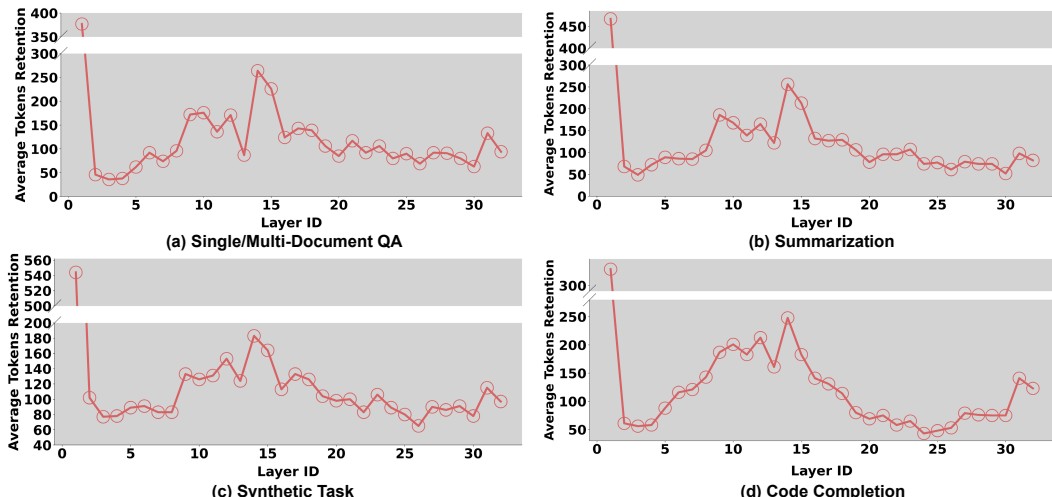

Figure 2: Average token retention across layers in LlaMA for different tasks, including (a) *Single/Multi-Document QA*, (b) *Summarization*, (c) *Synthetic Task*, and (d) *Code Completion*. There is a sharp decrease in token retention after the first layer, followed by varying patterns of fluctuation. Peaks are observed around Layer 15 and towards the final layers.

## 4 DYNAMICKV

During inference, the quadratic complexity of attention calculation results in a significant computational and memory burden, especially when processing long contexts. DynamicKV addresses this issue by focusing on inter-layer attention in large language models (LLMs), determining the appropriate size of KV cache to retain per layer through efficient awareness of inter-layer attention.

Rather than relying on a fixed retention pattern, such as pyramid-shape or average retention all layers, DynamicKV employs a progressive algorithm that dynamically adjusts token retention during the prefill phase. This dynamic retention strategy accelerates the decoding stage while maintaining minimal impact on overall memory usage.

Specifically, we first define layer $l \in \mathbb{R}^L$ and head $h \in \mathbb{R}^H$ in LLMs. For the calculation of attention scores, we use weights $W_Q \in \mathbb{R}^{N \times N}$, $W_K \in \mathbb{R}^{N \times N}$, and $W_V \in \mathbb{R}^{N \times N}$, with the input query embedding denoted as $X \in \mathbb{R}^{N \times M}$, $N$ is the dimension of the hidden size, and $M$ is the length of input tokens. Traditional token drop methods often consider the most recent tokens as the important ones for producing output information, as they retain relevant information needed for generating answers. We refer to these tokens collectively as the *current window*, with the window size denoted as $ws$. In the prefill phase, we adopt the method from Li et al. (2024), Zhang et al. (2024a), where the attention score is calculated by averaging over the current window and previous tokens, followed by pooling. The formula is as follows:

$$A_{l,h} = \text{pooling}(\frac{1}{ws} \sum_{i=1}^{ws} \text{Attention}(X_i, W_Q, W_K)), \tag{1}$$

here, pooling helps in understanding the context better and $A_{l,h}$ denotes the attention score for the l-th layer and h-th head. This approach allows us to effectively pool the attention scores, ensuring that key tokens are retained based on their relevance to both the current window and previous context.

Next, we set a fixed retention budget. Specifically, to ensure a fair comparison with other methods, we introduce the average retention length per layer, denoted as $wt$, and a scaling ratio, $r_{max}$. The calculation formula is as follows:

$$bs = (wt - ws) \times r_{max}, \tag{2}$$

here, $bs$ represents the size of retained KV cache across all layers. Next, we design a layer-aware progressive dynamic KV cache compression method. The prefill phase of LLMs involves a hierarchical forward process, where for each layer, we retain a KV cache of length $bs$ when computing

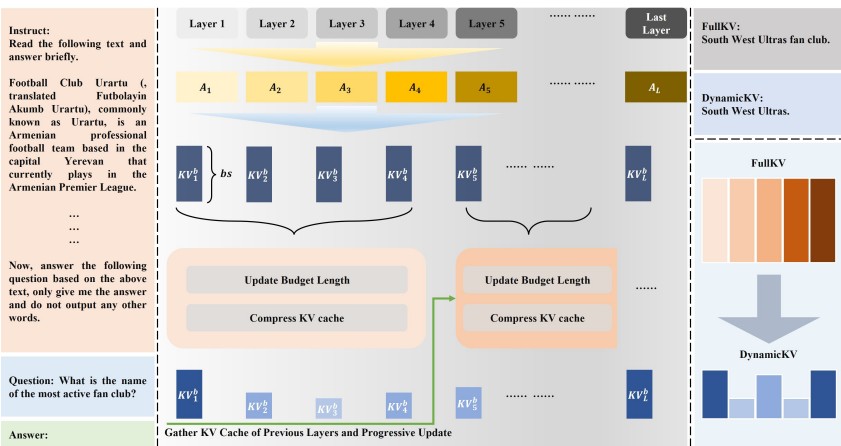

Figure 3: Overview of our DynamicKV structure and KV cache compression comparison. Left: input prompt, consisting of three parts: Instruct, Question, and Answer. Middle: DynamicKV structure, which progressively and dynamically updates the KV cache in stages to ensure that the total KV cache size remains within the maximum budget. Right: a comparison between DynamicKV and FullKV, highlighting the efficiency and resource savings achieved by our dynamic updating strategy.

$A$. Additionally, every $m$ layers, we perform an update across the current and all previous layers. Specifically, for each layer, we use a top-$k$ strategy to retain the largest $bs$ values from $A_l$, where $A_l$ represents the attention scores of layer $l$. The formula for this process is as follows:

$$A'_l = \text{TopK}(A_l, bs) \tag{3}$$

Next, we extract the indices in the original $A_l$ that correspond to the values in $A'_l$. The KV cache at these indices is retained as the compressed KV cache. Specifically, the retained KV cache is defined as:

$$\text{KV}'_l = \text{KV}_l[A'_l.\text{indices}] \tag{4}$$

where $A'_l.$indices represents the indices of the top-$k$ values in $A_l$. This ensures that the KV cache is compressed efficiently, retaining only the most important tokens for each layer while minimizing memory usage. To ensure that the memory required for hierarchical transmission remains small, the KV cache of each layer is initially compressed as described above. Every $m$ layers, we extract $A$ and perform a unified normalization across the completed layers, updating them layer by layer to ensure consistency across the entire hierarchy.

First, we fix the final size of KV cache to be retained, which is calculated as $(wt - ws) \times H \times l$, where $H$ is the number of heads and $l$ is the number of layers. Then, for each layer, the attention score $A$ is used to compute the length to retain for each layer $C_l$ via a top-$k$ strategy. The retention lengths for the first $m$ layers are then normalized to obtain a budget length $Z$, ensuring that the retention is distributed effectively across layers. The specific formula is as follows:

$$C_l = \text{Normlize}\left(\text{Count\_Elements}\left(\frac{\text{TopK}(A, (wt - ws) \times H \times l).\text{indices}}{(L \times M \times l)}\right)\right) \tag{5}$$

$$Z = \left[\frac{bs \times t}{\max(C_l)} \text{ for } t \in C_l\right] \tag{6}$$

$$r = \frac{\sum Z}{(wt - ws) \times L}, Z = \left[\frac{k}{r} \text{ for } k \in Z\right] \tag{7}$$

The KV cache is further updated layer by layer based on this normalized budget, progressively refining the retained information to align with the overall compression strategy. The above process can be expressed as Algorithm 1.

---

**Algorithm 1** DynamicKV in Prefill Phase

---

1: **Input:** initial budget K/V cache list $K^b$, $V^b$, radio max $r_{max}$, update interval $m$, mean token length $wt$, window size $ws$, sequence length $S$, head dimention $hd$, input embedding of window size $X^{ws} \in \mathbb{R}^{ws*d}$, initial budget Attention list computed by window token and others $A^b$,
2: **Output:** Compressed K/V cache $K^c$, $V^c$
3: $bs = (wt - ws) \times r_{max}$
4: **def** Update_Buffer_Length($A$, $l$):
5:     $A^{gather} \leftarrow$ cat((([$A$ for $l$ in (1, $l$)]), 0).view(-1)
6:     $cnts \leftarrow$ Count_Elemnets(topk($A^{gather}$, k=$(wt - ws) * H * l$).indices / ($L * S$)) / $l$
7:     Compute the $norm$ of $cnts$, range in (0, 1)
8:     $BL \leftarrow$ [int(($bs * t$ / max($norm$))) for $t$ in $norm$]
9:     $r \leftarrow$ sum($BL$) / (($wt - ws$)*$L$)
10:    $BL \leftarrow$ [int($k/r$) for $k$ in $BL$]
11:    Return $BL$
12: **for** $l \leftarrow 1$ **to** $L$ **do**
13:    Compute full KV states $K^s$, $V^s$
14:    **for** $h \leftarrow 1$ **to** $H$ **do**
15:        /* compute the Attention between window size token and other all token */
16:        $A_{l,h} \leftarrow$ softmax(($X^{ws}W_h^Q) \cdot K_h^T$).mean(dim=-2).pooling(dim=-1)
17:    **end for**
18:    Append $A_l$ to $A^b$ /* current $A_l$ shape is [H, S] */
19:    /* calculate current layer buffer KV cache */
20:    indices $\leftarrow A_l$.topk($bs$, dim=-1).indices.unsqueeze(-1).expand(-1, -1, $hd$)
21:    $K_l^b \leftarrow$ cat(($K^s$[:,:$-ws$,:].gather(dim=-2, indices),$K^s$[:,$-ws$:,:]), dim=-2)
22:    $V_l^b \leftarrow$ cat(($V^s$[:,:$-ws$,:].gather(dim=-2, indices),$V^s$[:,$-ws$:,:]), dim=-2)
23:    /* gradually compress*/
24:    **if** $l \% m$ == 0 **then**
25:        $Bl \leftarrow$ Update_Buffer_Length($A_l$, $l$)
26:        /* update the buffer K/V Cache*/
27:        **for** $i \leftarrow 1$ **to** $l$ **do**
28:            $K_i^b \leftarrow$ cat(($K_l^b$[:,:$Bl_i$,:], $K_l^b$[:,$-ws$:,:]), dim=-2)
29:            $V_i^b \leftarrow$ cat(($V_l^b$[:,:$Bl_i$,:], $V_l^b$[:,$-ws$:,:]), dim=-2)
30:        **end for**
31:    **end if**
32: **end for**
33: Update the K/V Cache $K^c$, $V^c$ from $K^b$, $V^b$

---

# 5 EXPERIMENTS

We conduct comprehensive comparative and ablation experiments to verify the effectiveness of our DynamicKV. In Section 5.1, we introduce the models, datasets and baselines used in our experiments. Section 5.2 provides a performance comparison between DynamicKV and baseline approaches. Next, in Section 5.3, we present the results of DynamicKV on the Needle in Haystack Task. Finally, in Section 5.4, we conduct an ablation study on the parameters of our method to validate its feasibility.

## 5.1 IMPLEMENTATION DETAILS

**Models and Context Length.** We utilize the official checkpoints of recently released models from huggingface including LlaMA-3-8B-Instruct(Dubey et al., 2024), Qwen-2-7B-Instruct(Yang et al., 2024), Mistral-7B-Instruct-v0.2(Jiang et al., 2023), and InternLM-2.5-7B-Chat-1M(Cai et al., 2024) as our base models, which support context lengths of 8k, 32k, 32k, and 1M tokens respectively.

**Datasets.** LongBench is a comprehensive benchmark for evaluating the contextual understanding capabilities of LLMs. For our comparative experiments, we use 16 English datasets from this benchmark, specifically NarrativeQA (Kočiský et al., 2018), Qasper (Dasigi et al., 2021), MultiFieldQA-en, HotpotQA (Yang et al., 2018), 2WikiMultihopQA (Ho et al., 2020), MuSiQue (Trivedi et al.,

Table 1: Performance comparison on the LongBench dataset for full KV cache, previous methods (StreamingLLM, H2O, SnapKV, PyramidKV), and our DynamicKV method, with KV cache sizes of 128 and 512, using models including LLaMA3-8B-Instruct, Mistral-7B-Instruct-v0.2, QWen2-7B-Instruct, and InternLM. Bold indicates the best performance.

| Model | Size | Method | Single-Document QA | | | Multi-Document QA | | | Summarization | | | Few-shot Learning | | | Synthetic | | Code | | Avg. |
|---|---|---|---|---|---|---|---|---|---|---|---|---|---|---|---|---|---|---|---|
| | | | NrtvQA | Qasper | MF-en | HotpotQA | 2WikiMQA | Musique | GovReport | QMSum | MultiNews | TREC | TriviaQA | SAMSum | PCount | PRe | Lcc | RB-P | |
| | | | 18409 | 3619 | 4559 | 9151 | 4887 | 11214 | 8734 | 10614 | 2113 | 5177 | 8209 | 6258 | 11141 | 9289 | 1235 | 4206 | – |
| LLaMA-3-8B-Instruct | – | FullKV | 25.16 | 31.81 | 39.59 | 43.09 | 36.15 | 21.77 | 28.62 | 23.34 | 26.33 | 75.00 | 90.50 | 42.36 | 5.20 | 69.25 | 59.04 | 53.93 | 41.95 |
| | 128 | StreamingLLM | 17.85 | 9.50 | 23.09 | 37.84 | 29.02 | 16.77 | 17.91 | 20.42 | 20.16 | 44.00 | 73.00 | 30.00 | 5.80 | 69.50 | 48.38 | 49.31 | 32.03 |
| | | H2O | 21.58 | 12.54 | 28.49 | 37.13 | 32.36 | 18.88 | 20.23 | 22.16 | 21.14 | 39.00 | 86.62 | 39.19 | 5.50 | 69.50 | 57.39 | 54.46 | 35.39 |
| | | SnapKV | 21.71 | 12.37 | 32.38 | 37.44 | 30.48 | 19.50 | 19.06 | 21.36 | 20.07 | 45.5 | 87.74 | 38.15 | 5.50 | 68.85 | 57.42 | 54.61 | 35.76 |
| | | PyramidKV | 22.26 | 16.65 | 30.73 | 38.97 | 29.28 | 19.19 | 19.92 | 22.06 | 20.87 | 68.00 | 88.95 | 38.23 | 5.92 | 69.50 | 57.20 | 51.54 | 37.45 |
| | | ours | 22.10 | 14.93 | 32.94 | 41.06 | 27.98 | 21.18 | 20.03 | 22.06 | 21.28 | 65.50 | 89.61 | 38.70 | 5.13 | 69.50 | 58.01 | 54.00 | **37.75** |
| | 512 | StreamingLLM | 19.03 | 12.78 | 28.67 | 37.83 | 29.97 | 16.55 | 20.30 | 20.94 | 24.56 | 61.00 | 75.43 | 30.82 | 5.86 | 69.50 | 51.93 | 49.98 | 34.70 |
| | | H2O | 22.84 | 16.80 | 32.36 | 41.43 | 34.07 | 19.30 | 22.28 | 22.81 | 23.69 | 41.00 | 90.46 | 40.19 | 5.54 | 69.50 | 57.52 | 55.43 | 37.20 |
| | | SnapKV | 24.62 | 22.78 | 37.88 | 42.96 | 34.82 | 20.65 | 22.63 | 22.54 | 23.93 | 70.00 | 90.39 | 40.30 | 5.74 | 69.50 | 60.27 | 55.85 | 40.30 |
| | | PyramidKV | 24.48 | 23.51 | 36.14 | 42.33 | 31.95 | 20.73 | 23.37 | 23.01 | 24.37 | 72.50 | 90.43 | 40.54 | 5.88 | 69.50 | 59.25 | 54.87 | 40.18 |
| | | ours | 24.78 | 24.76 | 36.84 | 44.13 | 33.25 | 20.82 | 23.00 | 22.76 | 24.14 | 72.50 | 90.39 | 40.76 | 5.78 | 69.50 | 61.40 | 56.91 | **40.73** |
| Mistral-7B-Instruct-v0.2 | – | FullKV | 26.63 | 32.99 | 49.34 | 42.77 | 27.35 | 18.77 | 32.87 | 24.24 | 27.10 | 71.00 | 86.23 | 42.96 | 2.75 | 86.98 | 56.93 | 54.49 | 42.71 |
| | 128 | StreamingLLM | 16.58 | 14.76 | 30.36 | 28.13 | 21.76 | 11.98 | 18.26 | 19.02 | 19.16 | 43.50 | 74.12 | 28.50 | 2.50 | 31.81 | 43.65 | 41.19 | 27.83 |
| | | H2O | 21.66 | 21.64 | 38.60 | 30.96 | 20.63 | 13.02 | 20.65 | 22.61 | 22.08 | 39.00 | 82.19 | 39.75 | 3.16 | 79.98 | 51.25 | 48.20 | 34.71 |
| | | SnapKV | 20.11 | 21.28 | 42.98 | 37.51 | 22.31 | 14.43 | 19.19 | 21.89 | 21.01 | 48.00 | 83.77 | 40.44 | 2.51 | 66.99 | 51.64 | 48.57 | 35.16 |
| | | PyramidKV | 22.11 | 22.52 | 43.04 | 33.57 | 22.98 | 15.69 | 20.56 | 22.52 | 21.36 | 65.50 | 83.84 | 40.03 | 2.89 | 67.26 | 51.51 | 46.42 | 36.36 |
| | | ours | 22.05 | 23.65 | 43.08 | 36.03 | 22.60 | 15.23 | 21.35 | 23.11 | 22.19 | 68.00 | 84.79 | 41.02 | 4.20 | 70.11 | 52.45 | 47.41 | **37.33** |
| | 512 | StreamingLLM | 19.05 | 17.21 | 36.82 | 30.64 | 21.84 | 10.56 | 24.47 | 19.84 | 25.48 | 62.00 | 72.82 | 29.49 | 2.71 | 46.15 | 42.55 | 30.06 | 30.06 |
| | | H2O | 22.33 | 25.75 | 44.09 | 32.76 | 22.88 | 14.96 | 23.53 | 22.96 | 24.53 | 41.50 | 85.53 | 41.54 | 3.39 | 86.20 | 55.11 | 50.81 | 37.37 |
| | | SnapKV | 24.95 | 27.97 | 49.04 | 39.93 | 25.18 | 17.64 | 24.14 | 23.69 | 24.47 | 67.50 | 86.04 | 41.14 | 2.90 | 86.98 | 56.73 | 53.11 | 40.71 |
| | | PyramidKV | 23.49 | 28.79 | 48.71 | 40.11 | 25.64 | 16.35 | 24.79 | 23.52 | 24.49 | 66.50 | 86.20 | 42.58 | 3.53 | 81.81 | 55.45 | 51.67 | 40.47 |
| | | ours | 25.63 | 29.11 | 48.41 | 39.85 | 26.62 | 16.72 | 24.73 | 23.72 | 24.83 | 70.50 | 86.74 | 43.01 | 3.20 | 83.57 | 55.40 | 52.35 | **40.90** |
| Qwen2-7B-Instruct | – | FullKV | 25.14 | 42.35 | 45.04 | 14.80 | 14.13 | 9.23 | 36.35 | 23.79 | 26.51 | 76.50 | 89.16 | 45.23 | 6.50 | 75.50 | 60.30 | 60.78 | 40.71 |
| | 128 | StreamingLLM | 19.25 | 23.63 | 26.51 | 14.00 | 15.30 | 7.46 | 18.07 | 19.30 | 18.30 | 47.00 | 77.92 | 31.57 | 6.50 | 17.00 | 42.52 | 41.94 | 26.64 |
| | | H2O | 20.33 | 30.43 | 34.22 | 13.61 | 13.37 | 7.81 | 20.72 | 21.66 | 18.44 | 40.00 | 86.94 | 42.17 | 7.00 | 70.50 | 53.45 | 53.76 | 33.40 |
| | | SnapKV | 22.26 | 31.62 | 38.95 | 16.05 | 17.71 | 7.66 | 18.91 | 21.41 | 18.21 | 46.00 | 87.61 | 42.01 | 6.50 | 63.50 | 54.87 | 53.03 | 34.14 |
| | | PyramidKV | 20.50 | 31.70 | 39.95 | 18.54 | 18.54 | 8.85 | 19.24 | 20.47 | 18.18 | 60.00 | 87.98 | 39.71 | 7.00 | 49.00 | 48.77 | 47.91 | 33.52 |
| | | ours | 22.77 | 35.57 | 42.62 | 14.80 | 16.35 | 8.31 | 21.41 | 21.97 | 19.56 | 58.00 | 88.18 | 40.93 | 6.50 | 70.00 | 53.58 | 52.50 | **35.82** |
| | 512 | StreamingLLM | 20.47 | 26.97 | 32.64 | 14.31 | 14.39 | 6.82 | 25.70 | 19.31 | 24.88 | 66.00 | 76.56 | 32.11 | 8.00 | 15.50 | 46.58 | 44.20 | 29.65 |
| | | H2O | 22.88 | 34.28 | 41.40 | 13.30 | 14.60 | 8.31 | 23.69 | 22.07 | 22.72 | 39.50 | 88.75 | 43.91 | 6.00 | 72.00 | 58.83 | 57.83 | 35.63 |
| | | SnapKV | 23.86 | 38.61 | 44.65 | 15.60 | 14.62 | 9.13 | 24.56 | 22.39 | 23.07 | 70.00 | 89.31 | 43.32 | 5.00 | 72.00 | 58.67 | 60.74 | 38.47 |
| | | PyramidKV | 24.47 | 37.60 | 43.51 | 14.48 | 12.83 | 8.99 | 23.59 | 22.30 | 22.41 | 74.00 | 89.21 | 43.40 | 6.50 | 74.00 | 57.67 | 56.14 | 38.19 |
| | | ours | 24.66 | 40.44 | 45.30 | 15.42 | 13.89 | 8.46 | 25.51 | 22.77 | 22.92 | 74.00 | 89.27 | 43.18 | 7.00 | 74.00 | 60.38 | 59.33 | **39.16** |
| InternLM-2.5-7B-Chat-1M | – | FullKV | 22.42 | 27.61 | 39.98 | 40.92 | 33.48 | 26.68 | 33.01 | 25.18 | 26.28 | 72.50 | 86.76 | 39.76 | 2.91 | 100.00 | 55.86 | 57.95 | 43.21 |
| | 128 | StreamingLLM | 17.91 | 13.02 | 24.31 | 24.27 | 16.01 | 11.29 | 17.29 | 20.62 | 18.06 | 48.5 | 67.53 | 21.93 | 0.82 | 87.39 | 43.45 | 42.79 | 29.70 |
| | | H2O | 16.16 | 17.17 | 27.94 | 26.83 | 17.83 | 17.81 | 13.99 | 22.59 | 16.9 | 39.50 | 81.87 | 32.15 | 1.32 | 96.50 | 48.30 | 47.27 | 32.79 |
| | | SnapKV | 19.65 | 17.44 | 35.29 | 27.36 | 18.58 | 19.79 | 12.76 | 22.42 | 16.31 | 48.00 | 80.23 | 31.35 | 0.95 | 95.00 | 49.47 | 48.22 | 33.93 |
| | | PyramidKV | 18.80 | 17.35 | 33.48 | 31.16 | 20.05 | 19.02 | 14.65 | 22.02 | 17.40 | 69.50 | 80.87 | 32.02 | 1.23 | 95.00 | 49.33 | 47.16 | 35.28 |
| | | ours | 17.93 | 19.89 | 34.15 | 31.50 | 19.03 | 20.60 | 15.14 | 22.41 | 18.15 | 70.00 | 83.09 | 32.44 | 0.86 | 95.50 | 49.33 | 47.16 | **36.07** |
| | 512 | StreamingLLM | 17.58 | 15.86 | 26.55 | 26.68 | 16.69 | 11.01 | 25.96 | 21.33 | 25.57 | 65.00 | 67.16 | 21.71 | 0.95 | 87.56 | 43.58 | 42.76 | 32.25 |
| | | H2O | 15.33 | 19.84 | 32.41 | 27.88 | 20.10 | 21.13 | 16.91 | 22.99 | 21.49 | 41.00 | 84.38 | 34.76 | 1.23 | 96.50 | 48.46 | 50.00 | 34.65 |
| | | SnapKV | 16.86 | 23.28 | 36.24 | 32.14 | 19.89 | 23.21 | 17.69 | 23.18 | 22.44 | 71.00 | 84.05 | 34.34 | 1.00 | 96.50 | 50.32 | 53.34 | 37.84 |
| | | PyramidKV | 17.62 | 21.08 | 37.52 | 32.21 | 21.31 | 22.03 | 19.37 | 24.06 | 22.22 | 73.00 | 83.94 | 34.61 | 1.05 | 95.50 | 50.45 | 49.72 | 37.86 |
| | | ours | 17.77 | 23.87 | 37.74 | 32.98 | 21.13 | 20.85 | 19.13 | 23.49 | 22.48 | 75.00 | 84.89 | 36.70 | 0.91 | 95.50 | 50.70 | 51.08 | **38.39** |

2022), GovReport (Huang et al., 2021), QMSum (Zhong et al., 2021), MultiNews (Fabbri et al., 2019), TREC (Li & Roth, 2002), TriviaQA (Joshi et al., 2017), SAMSum (Gliwa et al., 2019), PassageCount, PassageRetrieval-en, LCC (Guo et al., 2023), and RepoBench-P (Liu et al., 2023). These cover key long context application scenarios such as *Single-Document QA*, *Multi-Document QA*, *Summarization*, *Few-shot Learning*, *Synthetic Tasks*, and *Code Completion*. Additionally, for the experiment on Needle in Haystack task, we test the models across their maximum length ranges [8k, 32k, 1M] using the PaulGrahamEssays dataset.

**Baselines.** We evaluate recent fixed-pattern token dropping methods, including: (1) **StreamingLLM**, which utilizes attention sinks and rolling KV caches to retain the most recent tokens. (2) **H2O**, which employs a Heavy Hitter Oracle for KV cache eviction. (3) **SnapKV**, which selects important tokens for each attention head through clustering. (4) **PyramidKV**, which introduces a pyramid pattern where layers select important tokens in a monotonically decreasing manner.

## 5.2 COMPARATIVE EXPERIMENTS ON LONGBENCH

With the total KV cache size fixed at 128 and 512, we compare the performance retention of StreamingLLM, H2O, SnapKV, PyramidKV, and our proposed method, DynamicKV, relative to Ful-lKV. As shown in Table 1, DynamicKV demonstrates stable improvements even while maintaining an extremely low KV cache size relative to the total context (128: 1.7%; 512: 6.9%). Specifically, with the cache size of 128, DynamicKV outperforms the best alternative by 0.3%, 0.97%, 1.68%, and 0.79% on LLama, Mistral, Qwen, and InternLM, respectively, retaining 90%, 87%, 78%, and

83% of the overall performance. Moreover, with a cache size of 512, DynamicKV surpasses the highest-performing method by 0.43%, 0.19%, 0.69%, and 0.53% on the same models, retaining 97%, 96%, 96%, and 89% of FullKV's performance. The data in the table clearly demonstrate DynamicKV's effectiveness under extreme compression, achieving nearly FullKV-level performance with just 6.9% of the cache size. The experimental results show that DynamicKV can improve the effect of complex tasks such as *code completion* more obviously on the basis of maintaining PyramidKV performance, and greatly improve the performance upper limit of lower KV cache size.

## 5.3 VISUALIZATION ON NEEDLE-IN-HAYSTACK TASK

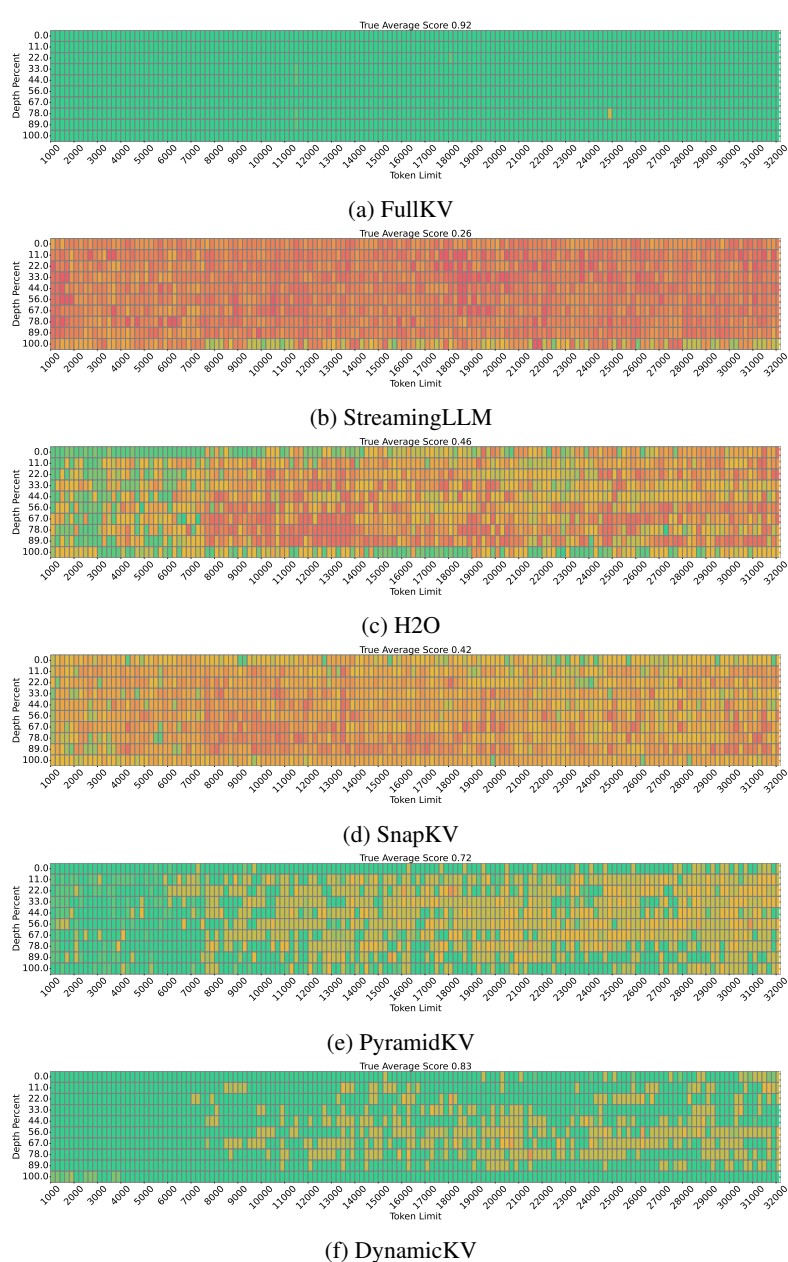

(a) FullKV

(b) StreamingLLM

(c) H2O

(d) SnapKV

(e) PyramidKV

(f) DynamicKV

Figure 4: Performance Comparison on the Needle in a Haystack Task Using Mistral-7B-Instruct-v0.2.

The needle-in-a-haystack test involves inserting key information at random positions within a long context and setting answers to evaluate whether LLMs can accurately detect critical information in

extensive contexts. To further illustrate the effectiveness of our approach in compressing the KV cache, we conduct additional experiments using Mistral on the needle-in-a-haystack task, focusing on maintaining an optimal size for the KV cache. As shown in Figure 4, we insert information at various positions in the Paul Graham Essays dataset and extract answers by prompting the model to generate responses. The green blocks indicate that the response matches the contents of the needle, but the color change from yellow to red indicates that the response is more irrelevant to the needle. We test a fixed KV cache size of 64 using FullKV, StreamingLLM, H2O, SnapKV, PyramidKV, and the DynamicKV method. The results indicate that DynamicKV maintains 90% of the model's performance even under extreme compression, improving accuracy by 57%, 37%, 41%, and 11% compare to the other methods, respectively. Additionally, the figure shows that with a context length of up to 7000, the extreme compression of DynamicKV nearly achieves full scores, and even beyond 7000, it shows significant improvements compared to other approaches. This finding illustrates that DynamicKV has a distinct advantage in hierarchical token selection and confirms that the number of critical tokens contained at different layers is always dynamic.

## 5.4 ABLATION STUDY

Table 2: Performance of DynamicKV with different KV cache size.

| KV size | LlaMA-3-8B-Instruct | Mistral-7B-Instruct-v0.2 | Qwen2-7B-Instruct | InternLM2.5-7B-Chat-1M |
|---------|---------------------|--------------------------|-------------------|------------------------|
| 64 | 34.93 | 33.95 | 32.67 | 33.67 |
| 96 | 36.70 | 36.22 | 34.85 | 35.31 |
| 128 | 37.75 | 37.33 | 35.82 | 36.07 |
| 256 | 39.83 | 39.23 | 36.98 | 37.29 |
| 512 | 40.73 | 40.90 | 39.16 | 38.39 |
| 1024 | 41.22 | 41.48 | 39.72 | 38.86 |

In this study, we investigate the performance of the DynamicKV mechanism across varying key-value cache sizes. The results, as shown in Table 2, reveal a consistent improvement in performance with an increase in the cache size for all evaluated models. For the Llama-3-8B-Instruct, the performance metric improved from 34.93 to 41.22 as the key-value cache size was increased from 64 to 1024. This improvement is also applicable to other models. These findings underscore the effectiveness of the DynamicKV cache in leveraging KV cache compression to maintain the capabilities of long context. Notably, a larger cache capacity is generally associated with superior performance. Nonetheless, it is essential to strike a balance when selecting the cache size, taking into account the practical constraints related to storage and computational resources.

## 6 CONCLUSION

In this study, we analyze the intrinsic patterns exhibited by large language models (LLMs) when processing long-context inputs across different task types. Our empirical findings reveal significant variations in the distribution of attention across these task types. Based on this observation, we introduce DynamicKV, a novel layer-aware KV cache compression approach that dynamically adjusts the KV cache size across layers. We evaluate the effectiveness and generalizability of DynamicKV through experiments on 16 datasets from the LongBench benchmark, demonstrating its broad applicability and performance benefits. From the results, we mainly conclude that: (1) a wave-like pattern is followed in complex reasoning tasks (e.g., *code completion* tasks); (2) a pyramid-like pattern is followed in *Synthetic* and *Summarization* tasks; (3) The dynamic hierarchical adaptive DynamicKV approach is capable of formulating a relatively appropriate KV cache retention strategy in accordance with diverse tasks. Particularly, in the circumstance of maintaining an extremely small KV cache size, the effect is significantly enhanced.; In the future, we hope that there is a more suitable method to perform KV cache compression without increasing the computation.

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

# A APPENDIX

## A.1 MODEL DETAILS

All the model structures and details in our experiment are shown in Table 3.

Table 3: Configuration of Models.

| Configuration | LlaMA-3-8B-Instruct | Mistral-7B-Instruct-v0.2 | Qwen2-7B-Instruct | InternLM2.5-7B-Chat-1M |
|---|---|---|---|---|
| Hidden Size | 4,096 | 4,096 | 3,584 | 4096 |
| # Layers | 32 | 32 | 28 | 32 |
| # Query Heads | 32 | 32 | 28 | 32 |
| # KV Heads | 8 | 8 | 4 | 8 |
| Head Size | 128 | 128 | 128 | 128 |
| Intermediate Size | 14,336 | 14,336 | 18,944 | 14336 |
| Embedding | False | False | False | False |
| Vocabulary Size | 128,256 | 32,000 | 151,646 | 92,544 |

## A.2 DATASET DETAILS

The data sources, average length, evaluation metrics, language, and data volume of the Long-Bench(Bai et al., 2023) dataset's subdatasets are shown in Table 4.

Table 4: An overview of the dataset statistics in LongBench.

| Dataset | Source | Avg len | Metric | Language | #data |
|---|---|---|---|---|---|
| *Single-Document QA* | | | | | |
| NarrativeQA | Literature, Film | 18,409 | F1 | English | 200 |
| Qasper | Science | 3,619 | F1 | English | 200 |
| MultiFieldQA-en | Multi-field | 4,559 | F1 | English | 150 |
| *Multi-Document QA* | | | | | |
| HotpotQA | Wikipedia | 9,151 | F1 | English | 200 |
| 2WikiMultihopQA | Wikipedia | 4,887 | F1 | English | 200 |
| MuSiQue | Wikipedia | 11,214 | F1 | English | 200 |
| *Summarization* | | | | | |
| GovReport | Government report | 8,734 | Rouge-L | English | 200 |
| QMSum | Meeting | 10,614 | Rouge-L | English | 200 |
| MultiNews | News | 2,113 | Rouge-L | English | 200 |
| *Few-shot Learning* | | | | | |
| TREC | Web question | 5,177 | Accuracy (CLS) | English | 200 |
| TriviaQA | Wikipedia, Web | 8,209 | F1 | English | 200 |
| SAMSum | Dialogue | 6,258 | Rouge-L | English | 200 |
| *Synthetic Task* | | | | | |
| PassageCount | Wikipedia | 11,141 | Accuracy (EM) | English | 200 |
| PassageRetrieval-en | Wikipedia | 9,289 | Accuracy (EM) | English | 200 |
| *Code Completion* | | | | | |
| LCC | Github | 1,235 | Edit Sim | Python/C#/Java | 500 |
| RepoBench-P | Github repository | 4,206 | Edit Sim | Python/Java | 500 |

