# OpenReview forum: "DynamicKV: Task-Aware Adaptive KV Cache Compression for Long Context LLMs"
_ICLR.cc/2025/Conference — Submitted to ICLR 2025_

### Official Review · Reviewer_JNWw · 2024-10-29

**Soundness:** 3
**Presentation:** 3
**Contribution:** 3
**Rating:** 5
**Confidence:** 4

**Summary:**

This paper addresses the challenge of managing the huge KV cache in LLMs for long-context tasks. Current KV cache compression methods use a fixed approach, often ignoring task-specific needs, which limits the retention of essential information. The authors introduce DynamicKV, a task-aware mechanism that adjusts KV cache sizes dynamically across layers based on task requirements. DynamicKV sets global and per-layer cache budgets, updating them periodically to optimize memory usage while retaining critical tokens.

**Strengths:**

The motivation is clear

The presentation is great. It is easy to read and follow

The empirical analysis is relatively comprehensive

**Weaknesses:**

The main weakness of this paper is there is no discusson on the benefits of dynamic KV.

- Did DynamicKV reduce the peak memory? Improve throughput? Reduce TTFT? Also, different budgets in different layers create load imbalance in pipeline parallelism, which is commonly used in LLM serving [1].

- Does this method compatible with commonly used serving framework and efficient techniques? Like PagedAttention [2] and quantization [3]

- What is the overhead of the Algo. 1 compared to the normal prefill phase?

- Is this framework compatible with Flashattention [4]? In long context scenario, FA is a must-use. In FA, you cannot explicitly obtain the full  attention score, as it is fused in FA

Also, please clearly state your setting of Needle test. Needle test is very sensitive to the prompt given to the model.


[1] Orca: A Distributed Serving System for Transformer-Based Generative Models

[2] Efficient Memory Management for Large Language Model Serving with PagedAttention

[3] Gear: an efficient kv cache compression recipe for near-lossless generative inference of llm

[4] FlashAttention-2: Faster Attention with Better Parallelism and Work Partitioning

**Questions:**

N/A

---

### Official Review · Reviewer_HMEi · 2024-10-29

**Soundness:** 2
**Presentation:** 2
**Contribution:** 2
**Rating:** 3
**Confidence:** 4

**Summary:**

This paper presents DynamicKV, a KV cache compression approach that dynamically adjusts the KV cache during the prefilling stage. It analyzes the distribution of the most important tokens across layers and task-aware distribution difference. It utilizes attention weights of recent tokens and all other tokens and updates the cache size of past layers every certain intervals. Experiments are conducted on LongBench and Needle-in-a-haystack across four LLMs.

**Strengths:**

- The authors analyze the token retention of different kinds of tasks across layers.
- Compared to compression with fix pattern, the proposed method can achieve higher compression ratios.

**Weaknesses:**

- The paper claims a task-aware approach, but the observation regarding token retention trends across different tasks appears to be quite consistent.
- The paper relies on scalar notation for its expressions, which does not accurately convey the potential vector or matrix operations intended.
- While the prefilling stage does incorporate a dynamic update mechanism for the cache every few layers, the budget determined remains static throughout the decoding phase.
- There is a concern about how the proposed allocation mechanism would work with batched inference. Different samples within the same batch may have varying attention patterns, which could lead to different budget requirements.

**Questions:**

- It is unclear where the boundary lies between task-aware and instance-aware adaptations. It is possible that different datasets within the same task category exhibit varying characteristics, which could account for the differing results observed in the Figure 2. Could you provide a token retention analysis for each dataset to clarify this distinction?
- There seems to be a misunderstanding in the interpretation of $A_l$ in Equation 3 and Algorithm 1. It should be a four-dimensional tensor with size `(batch_size, num_heads, win_len, seq_len)`. When applying TopK, $A_l$ should first be summed across the third dimension.
- Please provide a comparison of DynamicKV's performance against baselines in terms of efficiency metrics such as first token latency and memory consumption.

---

### Official Review · Reviewer_mnAq · 2024-11-03

**Soundness:** 3
**Presentation:** 2
**Contribution:** 2
**Rating:** 5
**Confidence:** 5

**Summary:**

The paper looks into the problem of accelerating LLM inference under long contexts. The key insights is that different tasks may need to allocate KV cache differently across layers. The paper then proposes a task-aware adaptive KV cache compression method, called DynamicKV. Evaluation on Longbench shows that the proposed method it is able to retain comparable accuracy with 6.9% tokens.

**Strengths:**

- The paper tackles an important and timely problem.
- Interesting observation that different tasks exhibit different KV cache patterns.
- Promising accuracy results under high compression ratio.

**Weaknesses:**

- The technical novelty is limited. Dynamic layer-wise KV cache allocation has been explored before, such as https://arxiv.org/pdf/2406.13035 and https://arxiv.org/pdf/2405.14366. It would be better if the authors can discuss and compare the proposed method with existing ones. Also, many of the optimizations proposed by this work are from prior studies, e.g., pooling from SnapKV and layer-wise allocation from pyramidKV.

- Significantly increased tuning cost. The method is not easy to use, it requires setting hyperparameters such as tw, ws, r-max per layer. For large LLMs that have close to hundreds of layers, this significantly increases the tuning overhead in addition to the fact that the method is task-specific, which means different tasks may need different tuning of those parameters. There are also no ablation studies on the sensitivity of the parameter choice.

- Missing comparison with related work on layer-wise adaptive KV.

- No measured system efficiency improvements. As a LLM inference optimization study, there is no real latency, throughput, and memory results reported in the paper. And it is unclear how the proposed method works with system optimizations such as FlashAttention, which reduces the primary bottleneck in long context inference, e.g., the attention map.

**Questions:**

The key formula, equation 2, does not seem to have any task-specific element in it, e.g., how to decide which task should use which allocation policy. It seems the algorithm introduces adaptive KV cache but leaves the real hard problem, e.g., how to figure out the optimal strategy for each task to users or to tuning?

---

### Official Review · Reviewer_D3mq · 2024-11-03

**Soundness:** 3
**Presentation:** 3
**Contribution:** 3
**Rating:** 6
**Confidence:** 4

**Summary:**

Efficient KV cache management is crucial for long-context tasks in LLMs, as extending context length increases memory demands. Current compression methods lack adaptability to specific tasks, leading to information loss. This paper presents *DynamicKV*, a task-aware mechanism that adjusts cache size per layer to optimize token retention based on task needs. Tested on the LongBench dataset, DynamicKV achieves high accuracy even with significant cache reduction.

**Strengths:**

- The experiment is solid
- The problem of finding more efficient architectures for transformers is relevant and not saturated

**Weaknesses:**

- The result of Needle in a Haystack is only on one model, maybe more models are better
- For the Needle in a Haystack task, it might be better to test longer with a model that can support a longer context window like InternLM-2.5-7B-Chat-1M or Llama-3-8B-Instruct-Gradient-1048k

**Questions:**

- How's your performance on longer context length and on different models?
- How's your performance on RULER?

---

### Official Review · Reviewer_nCAV · 2024-11-13

**Soundness:** 3
**Presentation:** 3
**Contribution:** 2
**Rating:** 3
**Confidence:** 4

**Summary:**

This work argues dynamically allocating kv cache budget across different layers to can result in better efficiency-performance trade off in long-context handling.

**Strengths:**

a. The argument of dynamic cache budget across different layers is sound and consistent with the findings of many recent literature.
b. Comprehensive model and baseline coverage for longbench evaluation.

**Weaknesses:**

a. Doesn't seem to support FlashAttention by the look of Eq 1.
b. No proper efficiency evaluation.
c. Dataset-wise, LongBench is too short to be utilized as the only long context evaluation. Please consider adding coverage of infinitybench and ruler, with a more long-context table model like llama 3.1/3.2.
d. I am interested in comparing DynamicKV with some newer head-based methods, such as Ada-KV and MInference.
e. The main argument of the DynamicKV is different tasks might prefer a different budget distribution for cache, but it looks like there is no significant distinction in Figure 2. Also, these four tasks are far from properly covering the diverse types of tasks a LLM will face. I'd like to see more variants. This request can be partially filled with (c) above, and some short context tasks can help, too.

I generally like this task-aware idea since works like Gear, InfLLM and many layer pruning works do seem to suggest different kv cache compression strategies can result in different performance across task types. However, I think the evaluation and practicality of this work can use a more comprehensive justification.

**Questions:**

a. What is the budget for Figure 4? It looks different to PyramidKV's needle results if it is 128.
b. Continuing on the diversity of tasks, I am not sure how DynamicKV will handle a change in task type during a multi-round conversation, or under batch inference where different requests might prefer different distribution patterns.

---

### Meta-Review · Area_Chair_q2p3 · 2024-12-21

**Metareview:**

This paper introduces DynamicKV, a task-aware KV cache compression method designed to enhance memory efficiency during inference for large language models (LLMs) by dynamically adapting cache sizes across layers based on task characteristics. While the method demonstrates promising accuracy retention under extreme compression on the LongBench dataset, the overall evaluation lacks breadth and depth. Specifically, the paper fails to address crucial aspects of efficiency, such as real-world latency and memory performance metrics. Additionally, the technical novelty of the method is limited, as key ideas overlap with existing techniques like SnapKV and PyramidKV. The task diversity in experiments is insufficient, and the use of LongBench alone is inadequate for a comprehensive evaluation of long-context tasks. Furthermore, the method's task-specific nature imposes a significant hyperparameter tuning overhead, which undermines its practical utility.

Despite the innovative idea of task-aware cache adaptation, the paper does not convincingly establish its benefits over existing methods in terms of practical gains, such as throughput or compatibility with efficient serving frameworks like FlashAttention. These gaps, along with limited novelty and incomplete evaluations, constitute the primary reasons for the decision to reject.

**Additional Comments On Reviewer Discussion:**

The reviewers consistently highlighted the lack of empirical evaluation on system efficiency and the limited task diversity in experiments. While the discussion acknowledged the potential of task-aware methods, the reviewers agreed that the paper does not provide sufficient evidence to justify its claims or to surpass the marginal acceptance threshold. Addressing these issues in future work could significantly strengthen the submission.

---

### Decision · Program_Chairs · 2025-01-22

Reject